# Continuous mapping of fine particulate matter (PM$_{2.5}$) air quality in East Asia at daily 6x6 km$^2$ resolution by application of a random forest algorithm to 2011-2019 GOCI geostationary satellite data

Drew C. Pendergrass[1], Daniel J. Jacob[1], Shixian Zhai[1], Jhoon Kim[2,3], Ja-Ho Koo[2], Seoyoung Lee[2], Minah Bae[4], Soontae Kim[4], and Hong Liao[5]

[1]School of Engineering and Applied Sciences, Harvard University, Cambridge, Mass., USA
[2]Department of Atmospheric Sciences, Yonsei University, Seoul, South Korea
[3]Particulate Matter Research Institute, Samsung Advanced Institute of Technology (SAIT), Suwon, South Korea
[4]Department of Environmental and Safety Engineering, Ajou University, Suwon., South Korea
[5]Jiangsu Key Laboratory of Atmospheric Environment Monitoring and Pollution Control, Jiangsu Collaborative Innovation Center of Atmospheric Environment and Equipment Technology, School of Environmental Science and Engineering, Nanjing University of Information Science and Technology, Nanjing, Jiangsu, China

*Correspondence to*: Drew Pendergrass (pendergrass@g.harvard.edu)

**Abstract.** We use 2011-2019 aerosol optical depth (AOD) observations from the Geostationary Ocean Color Imager (GOCI) instrument over East Asia to infer 24-h daily surface fine particulate matter (PM$_{2.5}$) concentrations at continuous 6x6 km$^2$ resolution over eastern China, South Korea, and Japan. This is done with a random forest (RF) algorithm applied to the gap-filled GOCI AODs and other data, including information encoded in GOCI AOD retrieval failure, and trained with PM$_{2.5}$ observations from the three national networks. The predicted 24-h GOCI PM$_{2.5}$ concentrations for sites entirely withheld from training in a ten-fold crossvalidation procedure correlate highly with network observations (R$^2$ = 0.89) with single-value precision of 26-32% depending on country. Prediction of annual mean values has R$^2$ = 0.96 and single-value precision of 12%. GOCI PM$_{2.5}$ is only moderately successful for diagnosing local exceedances of the National Ambient Air Quality Standard (NAAQS) because these exceedances are typically within the single-value precisions of the RF, and also because of RF smoothing of extreme PM$_{2.5}$ concentrations. The area-weighted and population-weighted trends of GOCI PM$_{2.5}$ concentrations for eastern China, South Korea, and Japan show steady 2015-2019 declines consistent with surface networks, but the surface networks in eastern China and South Korea underestimate population exposure. Further examination of GOCI PM$_{2.5}$ fields for South Korea identifies hotspots where surface network sites were initially lacking and shows 2015-2019 PM$_{2.5}$ decreases across the country except for flat concentrations in the Seoul metropolitan area. Inspection of monthly PM$_{2.5}$ time series in Beijing, Seoul, and Tokyo shows that the RF algorithm successfully captures observed seasonal variations of PM$_{2.5}$ even though AOD and PM$_{2.5}$ often have opposite seasonalities. Application of the RF algorithm to urban pollution episodes in Seoul and Beijing demonstrates high skill in reproducing the observed day-to-day variations in air quality as well as spatial patterns on the 6 km scale. Comparison to a CMAQ simulation for the Korean peninsula demonstrates the value of the continuous GOCI PM$_{2.5}$ fields for testing air quality models, including over North Korea where they offer a unique resource.

## 1. Introduction

Exposure to outdoor fine particulate matter (PM$_{2.5}$, less than 2.5 μm in diameter) is a global public health issue, accounting for 8.9 million deaths in 2015 [*Burnett et. al.,* 2018]. Beyond mortality, short-term exposure to elevated PM$_{2.5}$ levels is associated with numerous adverse health outcomes including increased hospital admissions for respiratory and cardiovascular issues [*Dominici et. al.*, 2006; *Wei et. al.*, 2019]. Long-term exposure is associated with neurodegenerative diseases such as dementia, Alzheimer's disease, and Parkinson's disease [*Kioumourtzoglou et. al.*, 2016]. High spatio-temporal monitoring of PM$_{2.5}$ concentrations to inform population exposure is important for both air quality regulation and epidemiological studies. Ground monitors can provide highly accurate measurements but have limited spatial coverage. Here we show how geostationary satellite observations of aerosol optical depth (AOD) over East Asia from the Geostationary Ocean Color Imager (GOCI) can be used with a random forest (RF) machine learning (ML) algorithm to provide continuous long-term reliable mapping of 24-h PM$_{2.5}$ at 6x6 km$^2$ spatial resolution.

The potential of satellites for high-resolution monitoring of PM$_{2.5}$ has long been recognized in the public health community [*Liu et al.*, 2004; *van Donkelaar et. al.*, 2006]. Satellites retrieve AOD by backscatter of solar radiation. The MODIS sensors launched in 1999 on the NASA Terra and Aqua satellites have been the main source of AOD data, with global coverage twice a day at up to 1 km resolution [*Remer et. al.*, 2005, 2013; *Lyapustin et. al.*, 2018]. Early approaches to relate AOD observations to surface PM$_{2.5}$ used chemical transport models (CTMs) to estimate local PM$_{2.5}$/AOD ratios [*Liu et al.,* 2004; *van Donkelaar et. al.*, 2006], with more recent studies adding ancillary satellite data on the vertical distribution of aerosol extinction [*Geng et. al.,* 2015*; van Donkelaar et. al.*, 2016; *van Donkelaar et. al.*, 2019]. Other approaches have used PM$_{2.5}$ network data to infer PM$_{2.5}$/AOD ratios [*Wang and Christopher*, 2003], with statistical models based on meteorological and land-use predictor variables to enable spatial extrapolation [*Gupta and Christopher*, 2009; *Liu et. al.*, 2009; *Kloog et. al.*, 2012; 2014].

More recently, non-parametric machine learning models have been developed to predict PM$_{2.5}$ from satellite AOD observations including neural networks [*Li et. al.,* 2017; *Zang et. al.*, 2019] and RFs, including approaches that fuse both [*Di et. al.,* 2019]. RF has been applied to MODIS AOD to produce high-resolution daily PM$_{2.5}$ products for the US [*Hu et. al.*, 2017] and China [*Guo et. al.*, 2021]. Others have used RF and satellite AODs to produce monthly PM$_{2.5}$ data over the North China Plain [*Huang et. al.*, 2018], as well as daily PM$_{2.5}$ data in California [*Geng et. al.*, 2020] and Cincinnati, Ohio [*Brokamp et. al.*, 2018].

Geostationary satellites are now dramatically increasing the capability for mapping of PM$_{2.5}$ from space. The GOCI instrument launched in 2010 by the Korea Aerospace Research Institute (KARI) observes AOD eight times daily at 0.5x0.5 km$^2$ pixel resolution over eastern China, the Korean peninsula, and Japan [*Choi et. al.*, 2018]. The fine-pixel hourly information is intrinsically valuable and also facilitates cloud clearing [*Remer et al.*, 2012]. GOCI AOD data aggregated to 6x6 km$^2$ resolution have been used to estimate PM$_{2.5}$ in regional studies for the Yangtze River Delta [*She et al.,* 2020] and eastern China [*Xu et al.*, 2015]. *Park et al*. [2019] find that PM$_{2.5}$ can be inferred over the Korean peninsula with greater accuracy using GOCI AOD than sparser MODIS data. AOD products from the Advanced Himawari Imager (AHI) onboard the Himawari-8 and -9 geostationary meteorological

satellites over East Asia have also been used to infer surface PM$_{2.5}$ [*Wang et. al.,* 2017; *Chen et. al.,* 2019].

AOD cannot be observed under cloudy conditions, and AOD retrievals from satellites can also fail for other reasons including snow surfaces. Different methods have been used to fill the data gaps and produce continuous data sets. Some studies use CTM AODs when satellite data are missing [*Hu et. al.,* 2017; *Stafoggia et. al.,* 2019]. *Kianian et. al.* [2021] used a statistical interpolation algorithm combining RF with the lattice kriging method to infer missing AOD over the US, while *Di et al.,* [2019] used a RF trained on gap-free covariates to fill in the gaps for MODIS AOD. Yet others first estimate PM$_{2.5}$ using available AOD observations, then infer missing PM$_{2.5}$ estimates using a separate gap-filling model [*Kloog et al.,* 2014; *She et al.,* 2020]. *Brokamp et al.* [2018] show that AOD missingness is itself predictive of PM$_{2.5}$, an insight we leverage in this work.

Here we apply a RF algorithm to 2011-2019 GOCI AOD data to construct a continuous dataset of 24-h PM$_{2.5}$ concentrations at 6x6 km$^2$ resolution for eastern China, South Korea, and Japan trained with surface network data. This is a larger spatial domain than has been attempted in previous studies. We ensure continuity by using gap-filled AOD, calculated by blending a CTM simulation with statistical interpolation, along with a parameter characterizing the length scale of the interpolation as inputs to the RF algorithm. This strategy maximizes training set size and allows the RF to determine a strategy to handle information encoded by retrieval failure. The resulting gap-filled product predicts PM$_{2.5}$ with comparable skill when AOD observations are absent as when they are available. We characterize the error in the RF-produced GOCI PM$_{2.5}$ dataset for both 24-h and annual concentrations and demonstrate the ability of the dataset to capture spatial and day-to-day variability on urban scales. We exploit the continuity of the dataset to determine trends of PM$_{2.5}$ air quality in East Asia over the past half decade.

## 2 Data and methods

### 2.1 Datasets

*GOCI AODs.* GOCI is onboard the Korean Communication, Ocean, and Meteorological Satellite (COMS) that was launched by KARI in June 2010 [*Choi et. al.,* 2012; *Choi et. al.,* 2016]. The first ocean color imager placed in geostationary orbit, GOCI covers a 2,500x2,500 km$^2$ domain centered on the Korean peninsula at 36ºN and 130ºE with 0.5x0.5 km$^2$ pixels observed every hour from 00:30 to 07:30 UTC. AOD at 550 nm over land is retrieved using the GOCI Yonsei aerosol retrieval (YAER) V2 algorithm at an aggregated 6x6 km$^2$ spatial resolution and 1 h temporal resolution [*Choi et. al.,* 2018]. Aggregation filters out pixels affected by sunglint or clouds, as well as the darkest 20% and brightest 40% pixels within the 6x6 km$^2$ scene [*Choi et. al.,* 2018]. We further aggregate the hourly AOD measurements of AOD into a daily mean for use in the RF.

Validation of the GOCI YAER V2 AOD with surface measurements from the AERONET surface network shows high correlation ($R = 0.91$), a root mean squared error (RMSE) of 0.16, and a mean bias (MB) of 0.01 with no significant spatial variation across East Asia [*Choi et. al.,* 2018]. GOCI YAER V2 also reports a Fine Mode Fraction (FMF) and a Multiple Prognostic Expected Error (MPEE) for the AOD but we find that they are not useful in our RF, as discussed later. For comparison, we also

calculate a RF trained on the GOCI-AHI fusion AOD product of *Lim et. al.* [2021]. The Advanced Himawari Imager (AHI) instruments onboard the Himawari-8 and -9 geostationary meteorological satellites were launched in October 2014 and November 2016, respectively. AHI has a larger field of view than GOCI but a shorter record.

*PM₂.₅ network data.* We use hourly $PM_{2.5}$ data from operational air quality networks in eastern China, South Korea, and Japan, and average them over 24 hours and over the 6x6 km$^2$ GOCI AOD grid to define targets for the RF algorithm. Data for eastern China are from the National Environmental Monitoring Center (https://quotsoft.net/air/) including 443 sites within the GOCI observing domain starting in May 2014 and increasing to 596 sites by 2019. Following *Zhai et. al.* [2019] we remove values with more than 24 consecutive repeats in the hourly timeseries as likely in error. Data for South Korea are from the AirKorea surface network of 123 sites (https://www.airkorea.or.kr/) starting in January 2015 and increasing to 298 sites by 2019. Data for Japan are from 1054 sites reported by the Japanese National Institute for Environmental Studies (NIES) for 2011-2017 (https://www.nies.go.jp/igreen/tj_down.html) and by the real-time Atmospheric Environmental Regional Observation System (AEROS) portal for 2018-2019 (Soramame; http://soramame.taiki.go.jp/DownLoad.php).

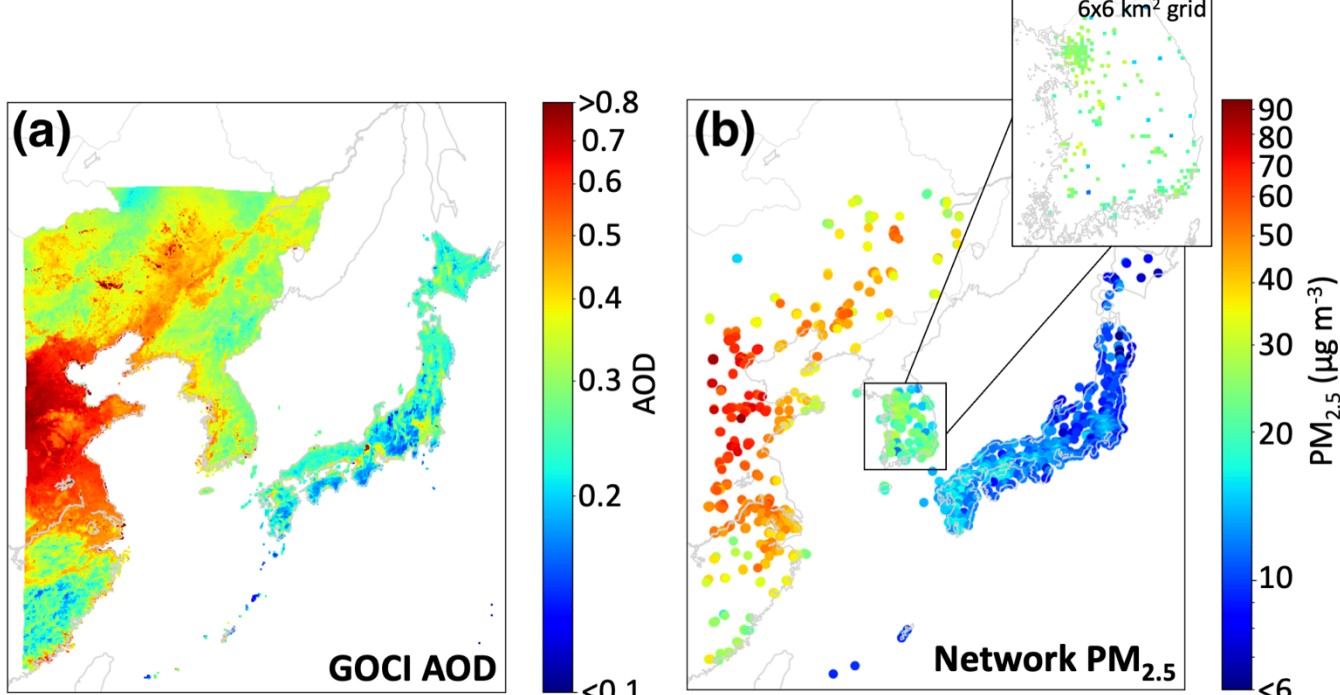

Figure 1: Mean aerosol optical depth (AOD) and surface network $PM_{2.5}$ concentrations over the Geostationary Ocean Color Imager (GOCI) viewing domain, 2011-2019. Panel (a) shows mean GOCI AOD data on the 6x6 km$^2$ grid. Panel (b) shows the mean surface network $PM_{2.5}$ data for eastern China (starting in May 2014), South Korea (starting in January 2015), and Japan, using large data symbols for visibility. Zoomed inset for South Korea shows the surface network observations with symbols corresponding to the 6x6 km$^2$ grid of the GOCI data. Log scale is used for colorbar.

*Meteorological and geographical predictor variables.* We use hourly meteorological data from the ERA5 global reanalysis, with resolution of 30x30 km$^2$ [*Hersbach et. al.*, 2020], as input predictor variables for the RF algorithm. For this purpose we aggregate the data to 24-h averages and allocate them to 6x6 km$^2$ GOCI grid cells by bilinear interpolation. We consider boundary layer height, 2-m air temperature and relative humidity (RH), 10-m meridional and zonal winds, and sea level pressure as potential meteorological predictor variables. We also include latitude, year, day of year (1-366), and nation category (eastern China, South Korea, or Japan) as geographical predictor variables. We considered 2015 population density [*CIESIN*, 2018] as a potential predictor variable but found that it was not useful as discussed in section 3.2.

**Figure 1** shows the mean distributions of GOCI AOD and surface network PM$_{2.5}$ for 2011-2019 or for the more limited durations of their records (2014-2019 for eastern China PM$_{2.5}$, 2015-2019 for South Korea PM$_{2.5}$). The PM$_{2.5}$ networks are extensive but coverage is nevertheless sparse and often limited to large urban areas, as illustrated by the zoomed inset for South Korea. We find that only 1.0% of GOCI 6x6 km$^2$ grid cells have PM$_{2.5}$ observations in eastern China, 7.4% in South Korea, and 7.9% in Japan. This geographic limitation in the PM$_{2.5}$ networks emphasizes the value of continuous coverage from the AOD data.

**2.2 AOD gap-filling**

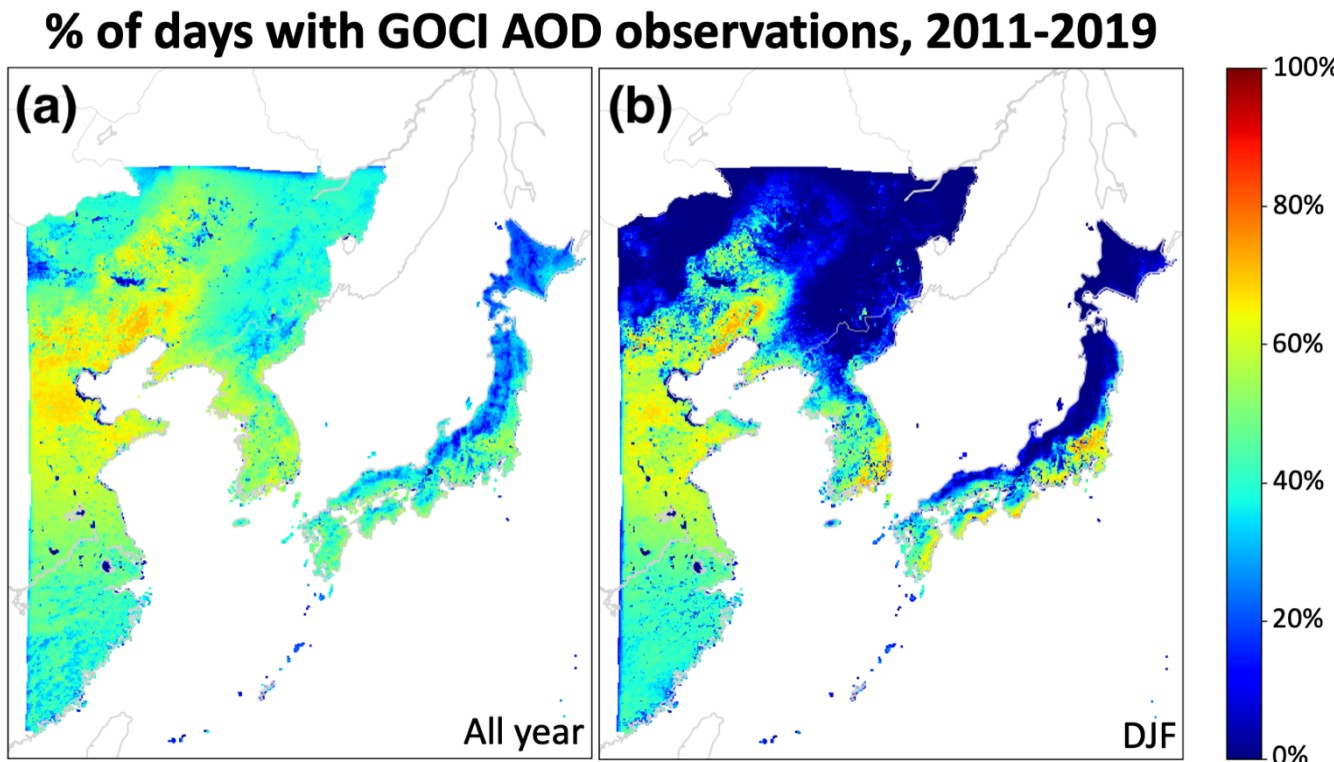

Figure 2: Percentage of days in 2011-2019 with at least one successful hourly retrieval of AOD on the 6x6 km$^2$ grid. Panel (a) shows year-round statistics while panel (b) shows winter months (DJF) only.

**Figure 2** shows the percentage of days with at least one successful hourly GOCI AOD retrieval on the 6x6 km$^2$ retrieval grid. There are substantial gaps in the record, mostly reflecting clouds and also snow cover in winter [*Choi et. al.*, 2018]. We seek to fill in these gaps to produce a continuous daily data set while accounting for the associated errors and leveraging information implicitly encoded in retrieval failure. We fuse two strategies according to the availability of nearby AOD retrievals: an inverse distance weighted (IDW) interpolation $AOD_{IDW}$ of nearby retrievals [*Shepard*, 1968] and a bias-corrected monthly $AOD_{GC}$ from the GEOS-Chem CTM:

$$AOD = \alpha \, AOD_{IDW} + (1 - \alpha)AOD_{GC} \tag{1}$$

where $\alpha$ is a weighting factor that depends on the distance from nearest retrievals. GEOS-Chem is a widely used CTM for inferring PM$_{2.5}$ from satellite AOD data [*Liu et al.*, 2004; *van Donkelaar et. al.*, 2006; 2016; 2019; *Geng et. al.*, 2015]. Here we use scaled monthly mean GEOS-Chem AODs from a simulation by *Zhai et al.* [2021] for 2016 in East Asia with 0.5°x 0.625° resolution, bias-corrected to the annual mean GOCI AODs on the 6x6 km$^2$ grid. In this way we obtain a spatial distribution of monthly mean $AOD_{GC}$ values for 2011-2019 for use in equation (1).

We calculate the weighting factors $\alpha$ used in Equation (1) via the Gaspari-Cohn function, a fifth-order piecewise polynomial with a radial argument $r$ [*Gaspari and Cohn*, 1999]. The Gaspari-Cohn function resembles a Gaussian distribution but with compact support, taking on a maximum value of 1 for $r = 0$ and a minimum value of 0 for $r \geq 2$. We define $r = l/c$ for a given 6x6 km$^2$ grid cell and day to be the distance $l$ from the midpoint of the grid cell to that of the nearest observed grid cell, normalized by a spatial correlation length scale $c$ determined from available AOD observations in and around that grid cell. We find that the value of $c$ ranges from 110 km to 170 km over our domain.

### 2.3 Random forest algorithm

**Table 1** lists the predictor variables included in the RF to infer 24-h PM$_{2.5}$ as dependent variable. RF is an ensemble machine learning method where many individual decision trees are fit to the training data and vote on an output value, with the average value taken as best estimate [*Breiman*, 2001].

**Table 1.** Random Forest predictor variables for 24-h PM$_{2.5}$[a]

GOCI gap-filled AOD observations[b]
    8-h average AOD at 550 nm wavelength
    $\alpha$ from Equation 1
Meteorology[c]
    Boundary layer height (m)
    10-m meridional wind (m s$^{-1}$)
    10-m zonal wind (m s$^{-1}$)
    2-m temperature (K)
    2-m relative humidity[d] (%)
    Sea-level pressure (Pa)
Metadata

Country dummy variables[e]
Latitude
Day of year
Year

[a]The RF algorithm predicts continuous 24-h PM$_{2.5}$ on a 6x6 km$^2$ grid for eastern China, South Korea, and Japan after training with PM$_{2.5}$ surface network data.

[b]8-hr average 550 nm AODs on the 6x6 km$^2$ grid retrieved with the YAER v2 algorithm [Choi et al., 2018]

[c] ECMWF ERA5 fields [*Hersbach et. al.,* 2020] at 30x30 km$^2$ spatial resolution and hourly temporal resolution, interpolated bilinearly to the GOCI grid and averaged over 24 hours.

[d] Estimated from temperature and dewpoint using the August-Roche-Magnus approximation [*Alduchov and Eskridge*, 1996].

[e]Three variables that, for each of eastern China, South Korea, and Japan, has value 1 if a grid cell is within those national borders and 0 otherwise.

Decision trees are fit recursively to the predictor variable. Suppose we have a collection of $N$ data elements $i \in [1, N]$, denoted $x_i$, each composed of $m$ predictor variables ($x_i \in \mathbb{R}^m$), and a corresponding list of $N$ labels $y_i$ that we would like to learn. In our case $y_i$ denotes the observed PM$_{2.5}$ concentrations from the surface networks averaged on the 6x6 km$^2$ grid, and $N$ denotes the number of these observations. The algorithm works by splitting the data into left and right subsets $L$ and $R$ at an optimum split point determined from the predictor variables in $x_i$ [*Pedregosa et. al.*, 2011]. The optimum split point is defined as the one that minimizes the impurity $G$,

$$G(L, R) = \beta \cdot \mathrm{MSE}(L) + (1 - \beta) \cdot \mathrm{MSE}(R) \tag{2}$$

where $\beta$ represents the fraction of data in the subset $L$ and MSE represents the mean squared error of each of the subsets,

$$\mathrm{MSE}(X) = \frac{1}{n} \sum_i (y_i - \bar{y})^2 \tag{3}$$

where $\bar{y}$ is the mean of the target labels within a given subset $X$ and $n$ is the number of elements in that subset. From there the same algorithm is recursively applied to the left and right subsets $L$ and $R$ until the tree is grown. We follow the advice of *Hastie et. al.* [2009] and grow trees until the data are fully classified (each leaf contains only one value).

Due to the recursive training structure, decision trees are sensitive to the data on which they are trained, because a change in one split point changes the composition of all its child nodes. Individual decision trees thus have high error variance but no inherent bias. It follows that averaging many individual and uncorrelated trees should yield a low variance, low bias prediction. We construct 200 trees in parallel and reduce correlation between them through a bagging procedure: for each of the 200 decision trees in the RF, sample the input data with replacement to form a new dataset of the same dimensions and then grow a decision tree from this bootstrapped data [*Breiman*, 2001]. Because of the high input sensitivity, a wide variety of decorrelated trees are grown. The predictions of each individual tree are averaged to yield the prediction of the RF. We fit our RF using the RandomForestRegression class in the Python module Scikit-learn [*Pedregosa et. al.*, 2011]. We attempted to further decorrelate trees by following *Breiman* [2001] and calculating split points of each individual tree using only a random subset of the $m$ predictor variables; however, a sensitivity test we performed showed only minor differences with the base case and therefore we follow *Guerts et. al.* [2006] in considering all predictor variables in the training process.

We evaluate how the RF generalizes to predictions for the full 6x6 km$^2$ domain via a 10-fold crossvalidation. For each fold of the crossvalidation, we leave out a randomly selected 10% of PM$_{2.5}$ network sites (averaged on the 6x6 km$^2$ grid if needed) from each country. These 10% represent the test set; because we perform the validation ten times, each grid cell is in the test set exactly once. We compare predicted PM$_{2.5}$ to withheld observed PM$_{2.5}$ using four metrics: root mean square error (RMSE); the RMSE divided by mean observed PM$_{2.5}$ (relative RMSE, or RRMSE); the coefficient of variation (R$^2$); and the mean bias computed by averaging the difference between predicted and observed PM$_{2.5}$ (MB).

An outcome of interest is the ability of our predictions to capture exceedances of National Ambient Air Quality Standards (NAAQS). We categorize each prediction within the test sets into one of four classes: true positives (TP) where both predicted and observed PM$_{2.5}$ exceed the NAAQS threshold; true negatives (TN) where neither exceed the threshold; false positives (FP) where an exceedance is predicted but not observed; and false negatives (FN) where an exceedance is observed but not predicted [*Brasseur and Jacob*, 2017; *Cusworth et. al.*, 2018]. We use these classes to compute three overall prediction grades. The first, percent of detection (POD), gives the fraction of observed exceedances that were successfully predicted:

$$POD = \frac{\Sigma\ TP}{\Sigma\ TP + \Sigma\ FN} \tag{4}$$

The second, false alarm ratio (FAR), gives the fraction of predicted exceedances that did not occur:

$$FAR = \frac{\Sigma\ FP}{\Sigma\ TP + \Sigma\ FP} \tag{5}$$

The third, equitable threat score (ETS), compares how well the prediction does relative to random chance:

$$ETS = \frac{\Sigma\ TP - \beta}{\Sigma\ TP + \Sigma\ FP + \Sigma\ FN - \beta} \tag{6}$$

where $\beta$ is the number of true positives obtained by random chance,

$$\beta = \frac{(\Sigma\ TP + \Sigma\ FP) \cdot (\Sigma\ TP + \Sigma\ FN)}{\Sigma\ TP + \Sigma\ TN + \Sigma\ FP + \Sigma\ FN} \tag{7}$$

ETS is 1 for perfect prediction skill and 0 for no better or worse than chance.

Predictor variable selection is an important task in implementing a RF, as the addition of non-informative variables can decrease performance. Unlike linear regression which can naturally ignore unhelpful predictors, irrelevant data can by chance aid in minimizing impurity $G$ at some stage in the

optimization process making all subsequent splits suboptimal. The six meteorological variables given in **Table 1** are standard in AOD/PM$_{2.5}$ prediction [e.g. *Kloog et. al.,* 2014; *Li et. al.*, 2017], while the four spatio-temporal variables (location dummies, latitude, year, and day of year) and the retrieval gapfilling parameter $\alpha$ proved to be informative in sensitivity tests. In addition to the predictor variables in **Table 1**, we considered as additional variables the population density, the GOCI fine mode fraction (FMF), and the GOCI multiple prognostic expected error (MPEE), but we found that they worsened accuracy of the fit and so we did not retain them. Because population density worsened the fit we did not include other spatially varying but temporally fixed land-use variables such as road data, elevation, or emissions. We also compared RFs trained on GOCI AOD and on GOCI-AHI fused AOD and found no significant difference in the fitting of PM$_{2.5}$. We therefore use the GOCI AOD product because of its longer record.

## 3 Results and discussion

### 3.1 Accuracy and precision of RF predictions

**Figure 3** shows scatterplots, color-coded by count, comparing surface observations of 24-h and annual mean PM$_{2.5}$ to the predicted GOCI PM$_{2.5}$ values in grid cells whose records are entirely withheld from training in the crossvalidation procedure. GOCI PM$_{2.5}$ values for the annual mean are obtained by averaging the 24-h predictions. **Table 2** gives comprehensive GOCI PM$_{2.5}$ evaluation statistics for East Asia and for each country. The 24-h predictions for East Asia have a negligible mean bias of 0.23 μg m$^{-3}$ (annual, 0.22 μg m$^{-3}$), though the RF underpredicts PM$_{2.5}$ at the high tail of the distribution; we will return to that issue later in the context of NAAQS exceedances. Root mean square error (RMSE) between observed and predicted 24-h PM$_{2.5}$ is 8.8 μg m$^{-3}$ (annual, 3.3 μg m$^{-3}$) corresponding to a relative RMSE (RRMSE) of 37% (annual, 14%), as defined in section 2.3. The prediction captures 89% of the observed 24-h variance ($R^2$ = 0.89) and 96% of annual ($R^2$ = 0.96). These results compare favorably to previous reconstructions of PM$_{2.5}$ from satellite AOD data. For example, a 1-km 2000-2015 continental US product and 3-km 2015-2016 east China product have crossvalidation $R^2$ of 0.86 and 0.87 respectively for daily PM$_{2.5}$ [*Di et. al.*, 2019; *Hu et. al.*, 2019], while a global 0.01° 1998–2018 product and a 0.1° degree 2000-2016 product for China have crossvalidated $R^2$ of 0.90-0.92 and 0.77 respectively for annual PM$_{2.5}$ [*Hammer et. al.*, 2020; *Xue et. al.*, 2019]. $R^2$ for annual mean PM$_{2.5}$ in South Korea is relatively low (0.41), which can be explained by the weak dynamic range of observed annual PM$_{2.5}$ in the country (**Figure 1**), as will be discussed later in this section.

Our gap-filling strategy does not introduce bias for days without GOCI observations (and with AOD inferred instead from equation (1)). **Figure S1** shows that surface network PM$_{2.5}$ has distinct distributions on days where AOD retrieval fails as compared to when AOD retrieval succeeds, a pattern successfully reproduced by GOCI PM$_{2.5}$. **Table 2** shows that the mean bias statistic on days where AOD retrieval fails is similar to the whole population. This suggests that the RF algorithm is able to successfully exploit the information encoded in AOD missingness in making a PM$_{2.5}$ prediction, a phenomenon also noted by *Brokamp et. al.* [2018].

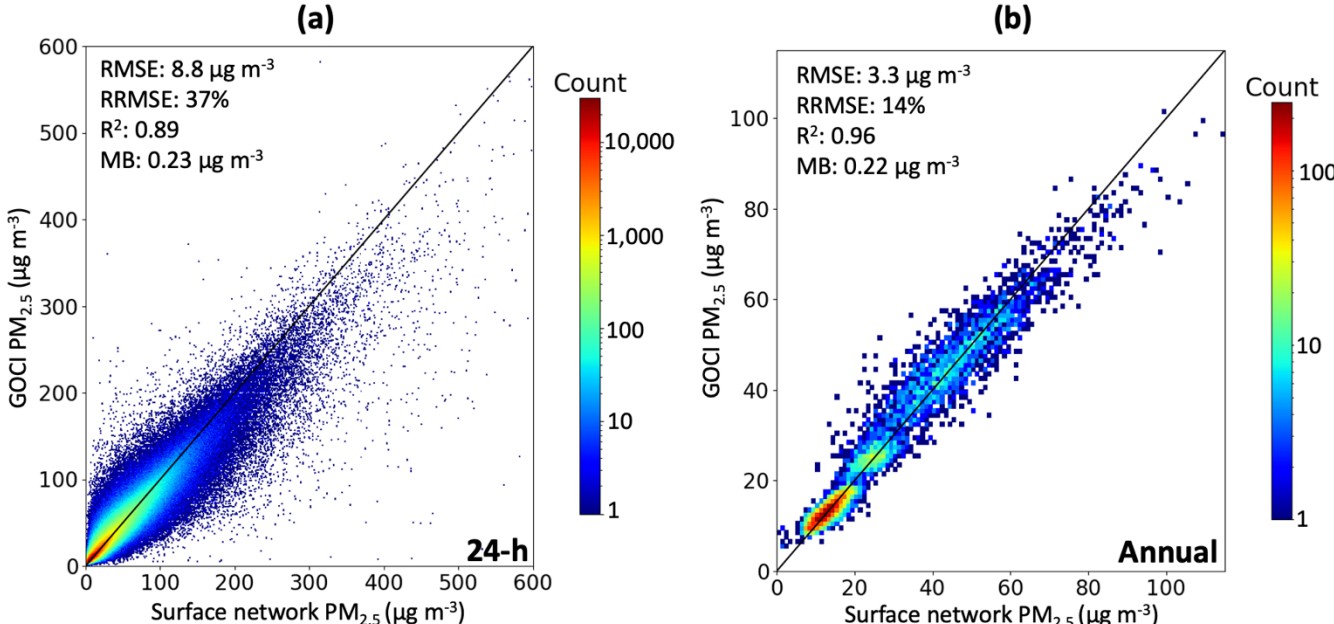

Figure 3: Ability of the random forest algorithm to predict 24-h (panel a) and annual mean PM$_{2.5}$ (panel b) in East Asia. Scatterplots depict the relationship between GOCI and surface network PM$_{2.5}$ at grid cells withheld from training in the crossvalidation. The plots are two-dimensional histograms where pixel color corresponds to the count of observation/prediction correspondences within the corresponding bin on a logged scale. The identity line is plotted in black. For annual mean PM$_{2.5}$, grid cells with fewer than 80% of PM$_{2.5}$ observation days in a given year are removed to avoid biasing the average. For panel (a), 0.002% of the data are not shown as they exceed the plot range; all data are shown in panel (b).

**Table 2.** Error statistics for fitting of PM$_{2.5}$ data by the RF algorithm[a]

|  | RMSE ($\mu g\ m^{-3}$) | RRMSE | $R^2$ | MB ($\mu g\ m^{-3}$) | MBnr ($\mu g\ m^{-3}$) |
|---|---|---|---|---|---|
| **24-h PM$_{2.5}$** |  |  |  |  |  |
| Overall | 8.8 | 37% | 0.89 | 0.23 | 0.23 |
| Eastern China | 15 | 32% | 0.85 | 0.49 | 0.53 |
| South Korea | 6.4 | 26% | 0.82 | 0.16 | 0.10 |
| Japan | 3.6 | 27% | 0.79 | 0.12 | 0.13 |
| **Annual PM$_{2.5}$** |  |  |  |  |  |
| Overall | 3.3 | 14% | 0.96 | 0.22 |  |
| Eastern China | 5.6 | 12% | 0.86 | 0.53 |  |
| South Korea | 2.9 | 12% | 0.41 | 0.24 |  |
| Japan | 1.6 | 12% | 0.70 | 0.094 |  |

[a]Comparison statistics between GOCI and surface network PM$_{2.5}$ are for the grid cells in each of eastern China, South Korea, and Japan completely withheld from the RF training process in the crossvalidation procedure. Statistics shown are for root-mean-square error (RMSE), relative RMSE (RRMSE), coefficient of variation ($R^2$), and mean bias (MB), and mean bias on days where AOD retrieval fails (MBnr).

One potential application of PM$_{2.5}$ monitoring from space would be to diagnose exceedances of national ambient air quality standards (NAAQS) at locations without network sites. **Table 3** shows the

NAAQS for 24-h and annual $PM_{2.5}$ for the three countries and the ability of GOCI $PM_{2.5}$ to diagnose NAAQS exceedances in grid cells excluded from the training process in the crossvalidation procedure. 24-h exceedances correspond to the high tails of the distributions but annual exceedances are much more widespread. The POD column shows percent of true positives successfully detected, while the FAR shows the rate of false positives (defined in section 2.3). POD for 24-h exceedances ranges from 47%-78% by country (FAR: 16%-21%). PODs are higher for annual exceedances but that reflects the higher observed frequency of these exceedances. The ETS values ranging from 0.43-0.63 indicate that the model captures exceedances with much better skill than random guessing.

**Table 3.** Ability of the RF algorithm to diagnose exceedances of air quality standards[a]

| | NAAQS ($\mu g\ m^{-3}$)[b] | Exceedance frequency[c] | | POD[d] | FAR[e] | ETS[f] |
|---|---|---|---|---|---|---|
| | | Observed | RF | | | |
| **24-h $PM_{2.5}$** | | | | | | |
| Eastern China | 75 | 16% | 15% | 78% | 16% | 0.63 |
| South Korea (old NAAQS) | 50 | 5.9% | 4.2% | 57% | 21% | 0.47 |
| South Korea (new NAAQS) | 35 | 19% | 17% | 73% | 20% | 0.55 |
| Japan | 35 | 1.6% | 0.91% | 47% | 17% | 0.43 |
| **Annual $PM_{2.5}$** | | | | | | |
| Eastern China | 35 | 77% | 83% | 97% | 9.2% | 0.54 |
| South Korea (old NAAQS) | 25 | 40% | 44% | 67% | 39% | 0.23 |
| South Korea (new NAAQS) | 15 | 100% | 100% | 100% | 0% | NA |
| Japan | 15 | 24% | 20% | 68% | 20% | 0.49 |

[a] Calculated using sites withheld from training in the crossvalidation procedure.
[b] National Ambient Air Quality Standards, specific to each country. We show results for the class 2 NAAQS in eastern China and for both pre-2018 ('old') and post-2018 ('new') NAAQS for South Korea because all observed grid cells exceed the new annual NAAQS of 15 $\mu g\ m^{-3}$.
[c] Percentage of site-days (24-h standard) or site years (annual standard) exceeding the NAAQS.
[d] Percent of detection (POD) defined as the percentage of exceedances successfully detected.
[e] False alarm ratio (FAR) defined as the percentage of predicted exceedances that did not occur.
[f] Equitable threat score (ETS) defined as the ability of the RF to predict exceedances beyond random chance.

The main difficulty for GOCI $PM_{2.5}$ to predict NAAQS exceedances is that many of those exceedances fall within the precision of individual predictions. This is illustrated in **Figure 4** with the cumulative probability density function (pdf) of the 24-h and annual mean $PM_{2.5}$ concentrations in eastern China, South Korea, and Japan, representing the same withheld data from the crossvalidation as in **Tables 2** and **3**. The 24-h RRMSE of 26-32% depending on country (**Table 2**) is shown as the grey envelope and is relatively flat across the distribution. Prediction of NAAQS exceedances within that uncertainty envelope is limited by the precision of the algorithm. All of the 24-h exceedances in Japan are within that envelope, as are most of the exceedances in eastern China and Korea. China has the largest fraction of exceedances beyond the RRMSE of the GOCI $PM_{2.5}$ and therefore the best prediction success. An additional though smaller cause of bias is that GOCI $PM_{2.5}$ underestimates the high tail of the pdf, as is apparent in **Figure 4**, which explains in particular why we achieve a better FAR than POD for 24-h $PM_{2.5}$ in South Korea and Japan. Our worst NAAQS prediction performance is for annual $PM_{2.5}$ in South Korea for the old 25 $\mu g\ m^{-3}$ standard, because most of the distribution is within the

RRMSE envelope. Additionally, the already small dynamic range of surface network annual PM$_{2.5}$ (black dots) is underestimated by the GOCI PM$_{2.5}$ (blue dots). These culminate in a GOCI PM$_{2.5}$ estimate with good RMSE but low $R^2$.

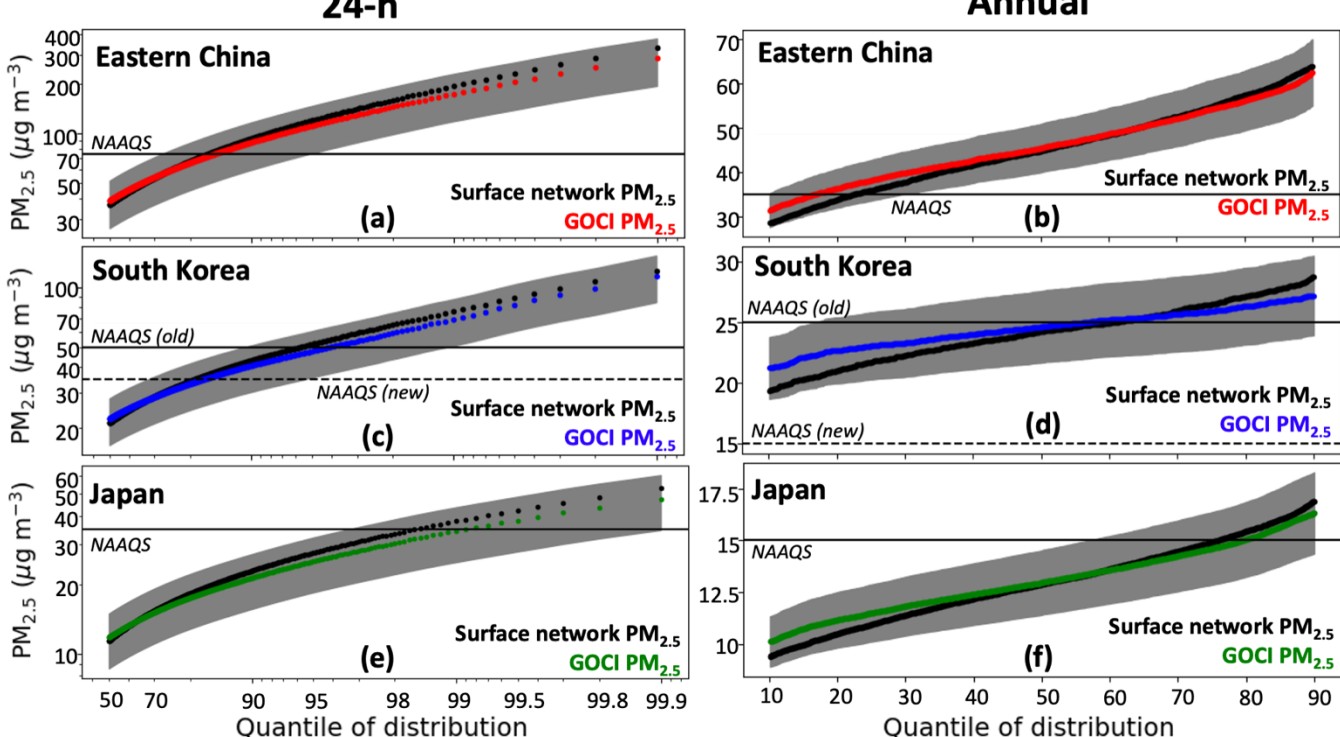

Figure 4: Cumulative probability density functions (pdfs) of 24-h and annual mean PM$_{2.5}$ concentrations in Eastern China, South Korea, and Japan. Surface network PM$_{2.5}$ (black) is compared to GOCI PM$_{2.5}$ (colored) taken from the crossvalidation. The grey envelope represents the relative root mean square error (RRMSE) of the RF algorithm as given in Table 2, measuring the predictive capability of the algorithm for individual events. The NAAQS for each country is shown as the horizontal line, with both the pre-2018 and post-2018 NAAQS shown for South Korea. Left panel scales are log-log while right-panel scales are linear. y-axis scales vary for the different countries.

We experimented with several modifications to the RF algorithm to improve prediction of NAAQS exceedances but with no success. These tests included training separate RFs for each of the three countries; training annual PM$_{2.5}$ predictions on annual (rather than 24-h) PM$_{2.5}$ data; directly predicting NAAQS exceedances by setting the learned label to be true if a day (year) is above the 24-h (annual) NAAQS for a given country; and applying different weights to the data so that the high tail is oversampled in the training process. None of these tests yielded significant improvements. Smoothing of the tails in RFs is a well-recognized problem [*Zhang and Lu*, 2012]. Following *Zhang and Lu* [2012] we attempted to train RFs to predict and correct the residuals but found this to be ineffective. Part of this tail smoothing could also result from the underlying GOCI AOD land product, which has a negative bias (-0.02) for high AODs and a positive bias (+0.02) for low AODs [*Choi et. al.*, 2018].

### 3.2 PM2.5 temporal trends and spatial distributions

**Figure 5** shows long-term trends of annual $PM_{2.5}$ for each country, as measured by the $PM_{2.5}$ surface network and as inferred in the GOCI $PM_{2.5}$ for both areal and population-weighted means. We do not include GOCI $PM_{2.5}$ for years before the networks became available (and hence when the RF could be trained) because of concern over extrapolation bias. The $PM_{2.5}$ networks show decreasing trends in all three countries and these trends are consistent with the GOCI $PM_{2.5}$ for both areal and population-weighted means, demonstrating that the trends reported by the $PM_{2.5}$ networks are representative of the countries. However, the $PM_{2.5}$ networks in eastern China and South Korea underestimate the population-weighted means. Trends in South Korea and eastern China become flat between 2018 and 2019 (with a slight population-weighted increase in South Korea). This could possibly reflect interannual meteorological variability [*Zhai et al.*, 2019; *Koo et. al.*, 2020], but also an increase in oxidants producing secondary aerosol [*Huang et. al.*, 2021]. **Figure S2** shows maps of annual GOCI $PM_{2.5}$ across the entire study domain.

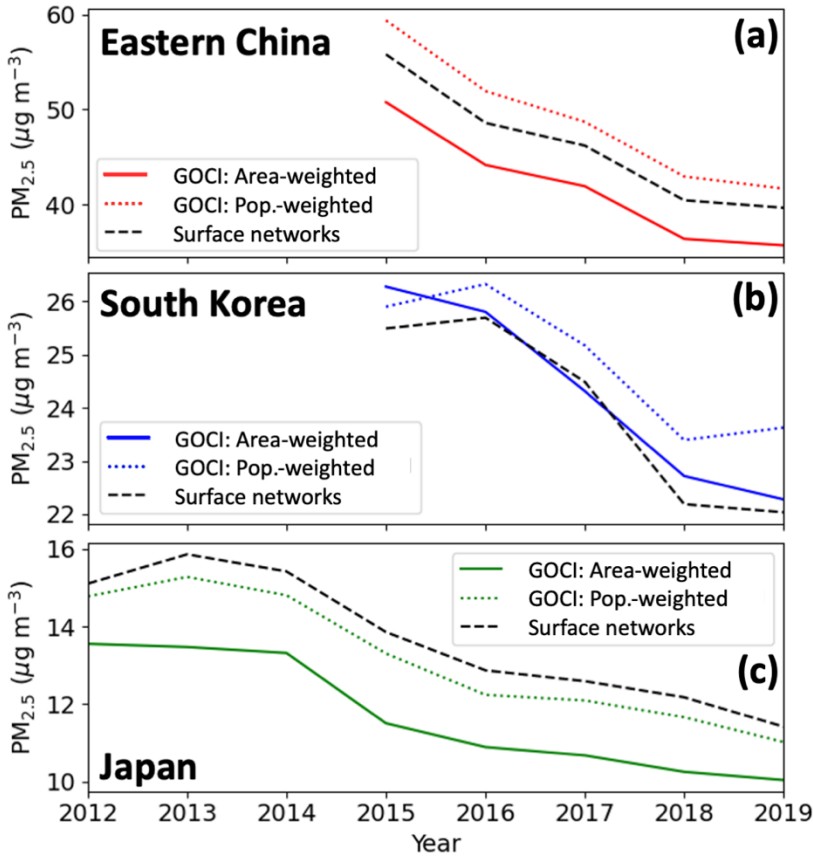

Figure 5: Trends in annual mean $PM_{2.5}$ concentrations for eastern China, South Korea, and Japan. Trends determined from the national surface $PM_{2.5}$ networks (dashed black line) averaged over 6x6 $km^2$ grid cells, requiring at least 80% of data for all years plotted, are compared to GOCI $PM_{2.5}$ trends inferred by the random forest (RF) algorithm with continuous temporal and spatial coverage on the 6x6 $km^2$ grid and weighted either by area (solid colored line) or by population (dashed colored line). Here we use an RF trained on all the data. Gridded

population data are from CIESIN [2018]. The national $PM_{2.5}$ networks include 413 continuously observed grid cells in eastern China, 74 in South Korea, and 307 in Japan. Trends are initialized at the onset of the surface network for complete years of data; due to the unavailability of the early months of the year, 2011 is discarded for Japan and 2014 for eastern China.

**Figure 6** shows the changes in annual mean $PM_{2.5}$ concentrations over South Korea between 2015 and 2019 as observed from the national network and as inferred from GOCI. We focus on South Korea here because it demonstrates how GOCI $PM_{2.5}$ adds considerable information to a region that already has relatively good network coverage, including detection of $PM_{2.5}$ hotspots missing from the network such as the Iksan region on the west coast in 2015 that was subsequently added to the network by 2019. **Figures S3** and **S4** show analogous maps for China and Japan, respectively.

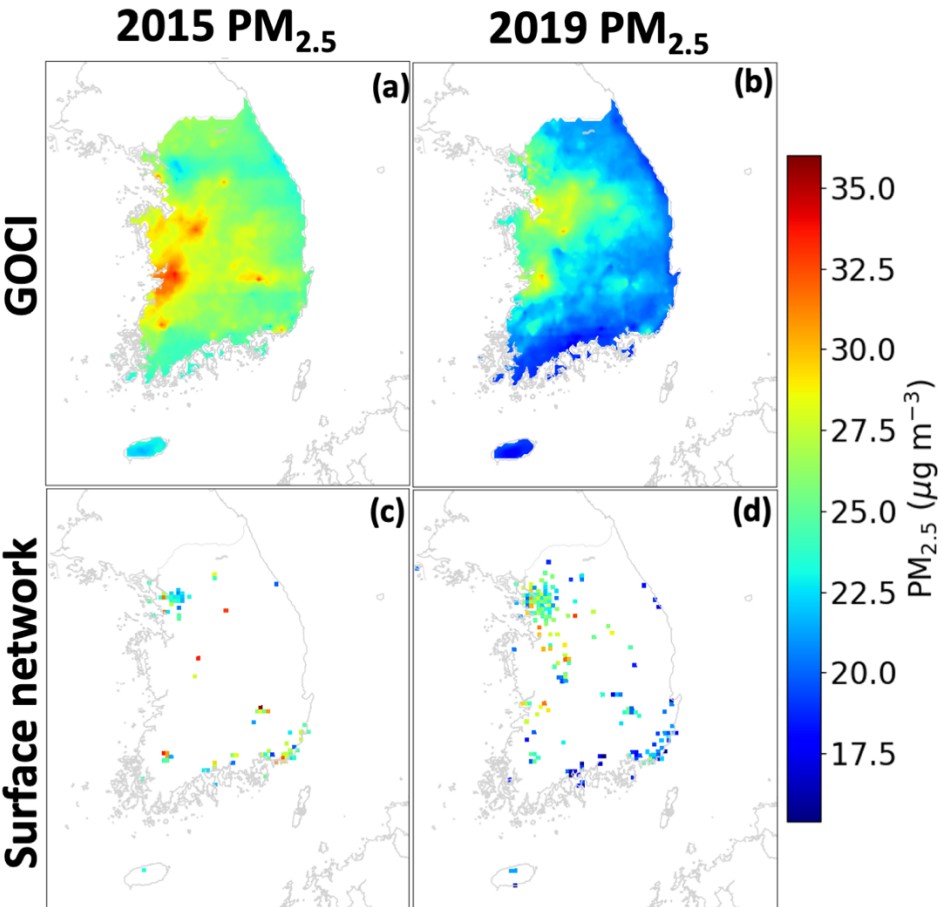

Figure 6: Annual mean $PM_{2.5}$ concentrations in South Korea in 2015 and 2019. GOCI $PM_{2.5}$ (top) inferred from an RF trained on all available data are compared to AirKorea network observations (bottom). Network observations are shown only if at least 80% of the year was observed.

**Figure 7** depicts the relative 2015-2019 trends of $PM_{2.5}$ concentrations in South Korea derived from a linear regression applied to the annual GOCI $PM_{2.5}$ in each 6x6 km$^2$ grid cell. Such a spatially resolved trend analysis is uniquely enabled by the GOCI coverage. We find decreases across the country except in the Seoul Metropolitan area which mostly shows no significant trend except for a few

pixels in Incheon. These results are consistent with the spatial patterns calculated from AirKorea data by *Yeo and Kim* [2019], who found 2015-2018 decreases in Incheon but not Seoul or the surrounding Gyeonggi province. Despite the insignificant changes in Seoul, substantial PM$_{2.5}$ decreases are found over other large urban areas including Busan, Ulsan, Daegu, and Gwangju. The three rapidly decreasing spots on the southern coast are Gwangyang, Sacheon, and Changwon, which house industrial

complexes related to the South Korean shipbuilding industry that has recently declined [*Jung-a* 2016]. **Figure S5** shows absolute 2015-2019 trends of GOCI PM$_{2.5}$ concentrations across the entire study domain, and demonstrates that the North China Plain has the largest overall PM2.5 reductions.

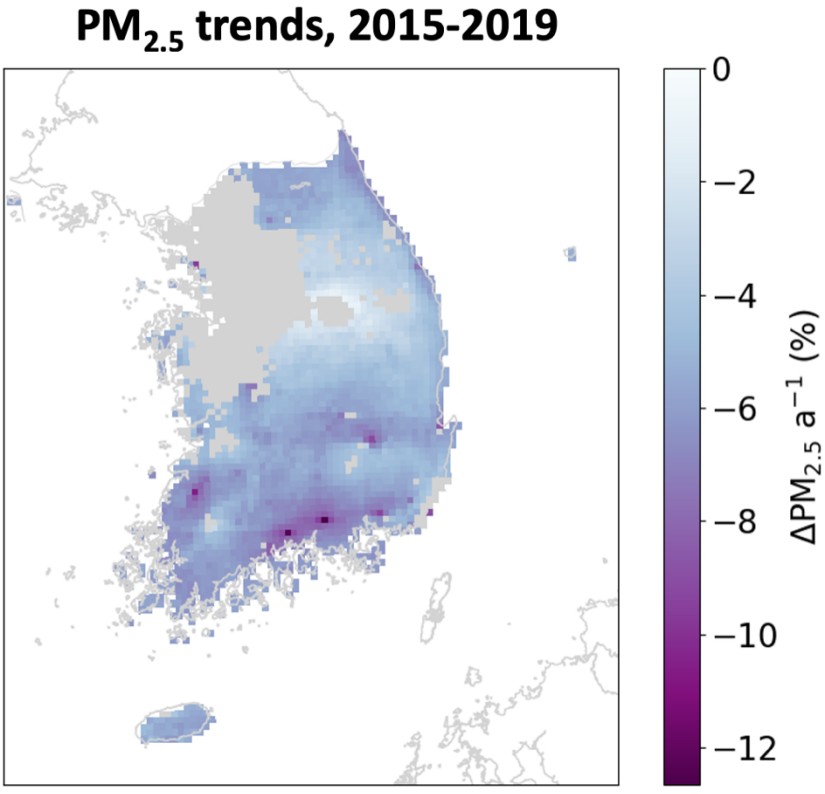

Figure 7: 2015-2019 trends per year in PM$_{2.5}$ concentrations across South Korea. The trends are obtained by ordinary linear regression of the annual mean GOCI PM$_{2.5}$ in each 6x6 km$^2$ grid cell with significant regression slopes ($p < 0.05$), where the RF is trained on all the available data. Grid cells with insignificant trends are plotted in gray.

        AOD and PM$_{2.5}$ in East Asia tend to have opposite seasonalities driven by boundary layer depth and RH [*Zhai et al.*, 2021]. **Figure 8** compares GOCI and surface network monthly mean PM$_{2.5}$ in the

Beijing, Seoul, and Tokyo metropolitan areas, with predictions coming from withheld data in the 10-fold crossvalidation. Correspondence between GOCI and network PM$_{2.5}$ may be tighter than the nationwide annual means plotted in **Figure 5** because these urban areas are well-observed. We see that the RF algorithm fully captures the observed seasonality in PM$_{2.5}$, although some observed monthly spikes are underestimated. The Figure illustrates the lack of trend in the Seoul Metropolitan Area over

2015-2019 but also shows that winter and summer PM$_{2.5}$ in the region have opposite and roughly equal trends, with winter growing more polluted while summers become cleaner.

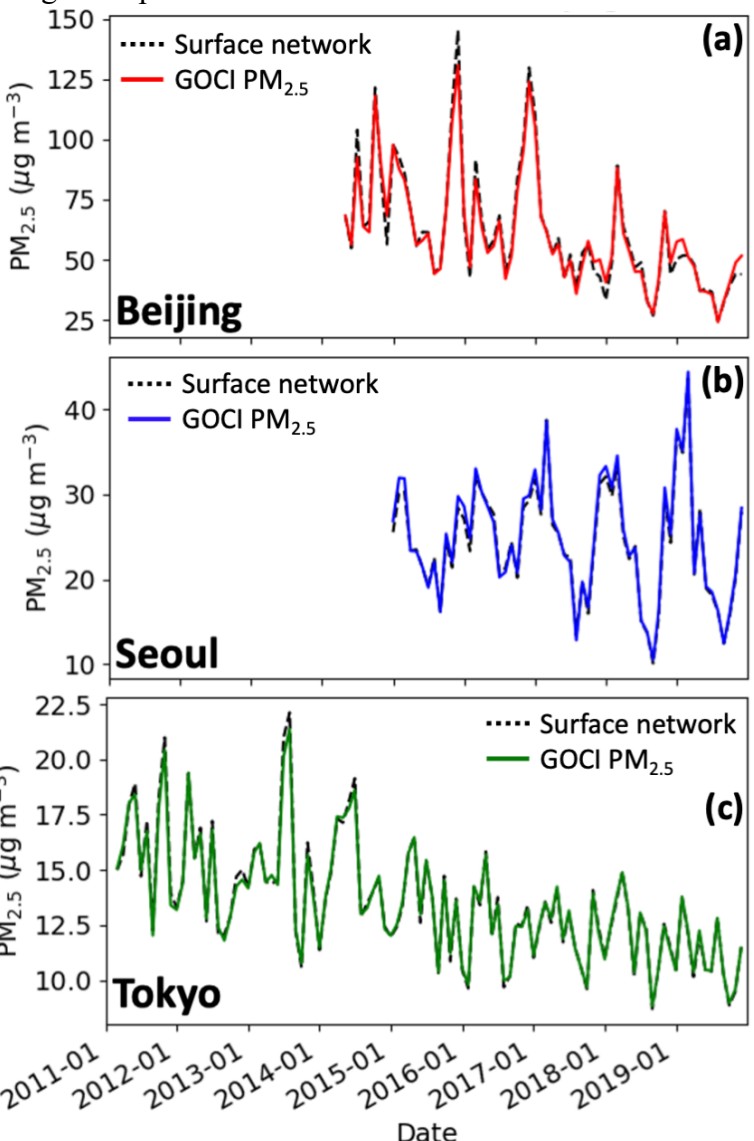

Figure 8: Monthly PM$_{2.5}$ concentrations in the Beijing Seoul and Tokyo metropolitan areas. GOCI PM$_{2.5}$ inferred from the RF algorithm for totally withheld sites in the crossvalidation are compared to network observations. Beijing is defined by the namesake province
boundary, Seoul by the Seoul and Incheon boundaries, and Tokyo as Ibaraki, Saitama, Chiba, Tokyo, Kanagawa, and Yamanashi prefectures.

### 3.3 Urban-scale pollution events

We examine here the ability of GOCI PM$_{2.5}$ to capture the spatial and temporal variability of PM$_{2.5}$ pollution events on urban scales. **Figure 9** shows a map of GOCI PM$_{2.5}$ — produced by a RF
trained on all the data, with surface network PM$_{2.5}$ overlaid — across the Seoul metropolitan area on

May 24-29, 2016 corresponding to a severe pollution event sampled during the KORUS-AQ field campaign [*Crawford et. al.*, 2021]. The dense PM$_{2.5}$ network for Seoul shows large variability at the sub 6x6 km$^2$ scale that GOCI does not resolve. However, GOCI PM$_{2.5}$ captures most of the variability in the network data aggregated on the 6x6 km$^2$ grid (R$^2$ = 0.74). It also captures successfully the day-to-day variability during the event.

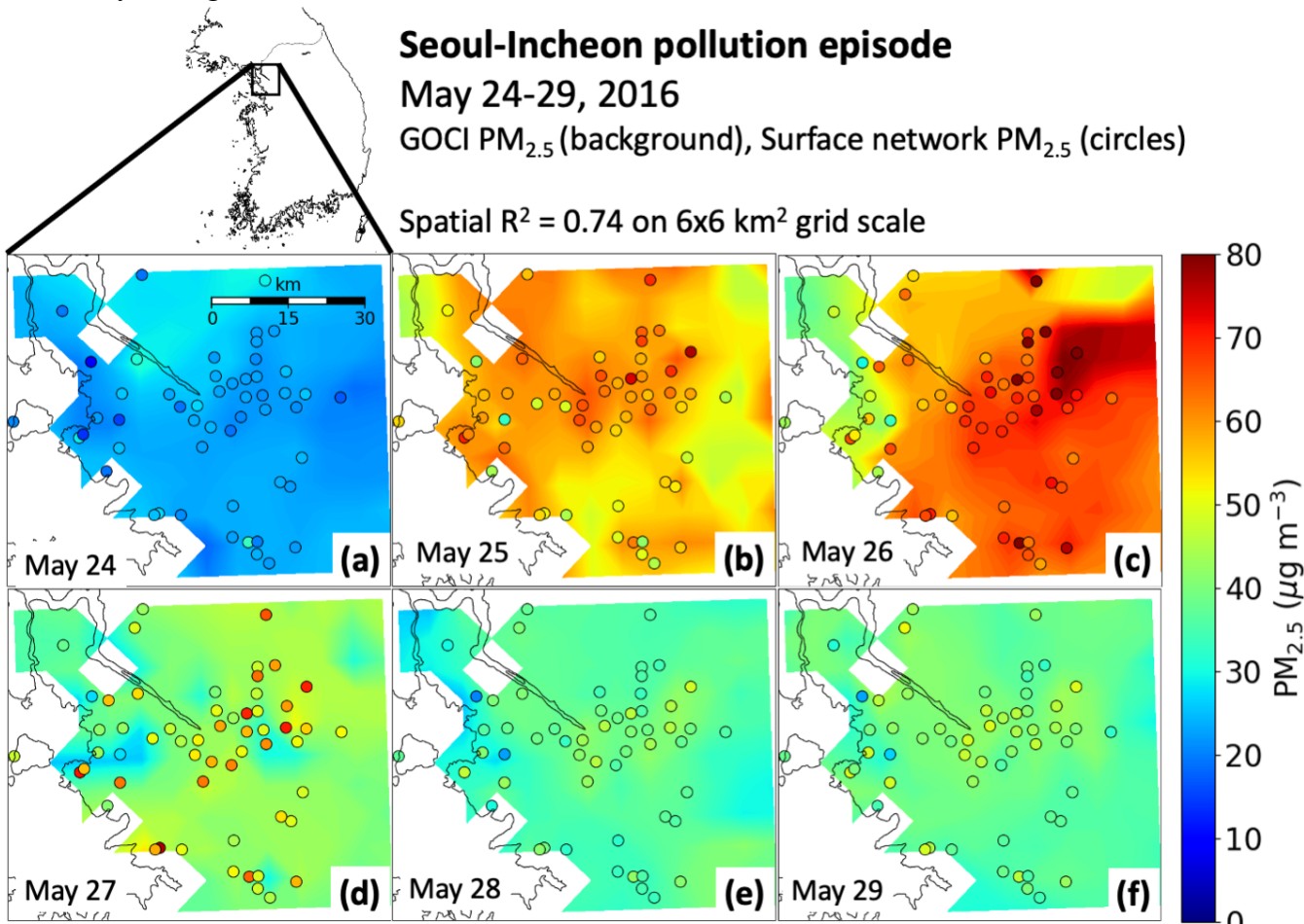

Figure 9: 24-h PM$_{2.5}$ concentrations during a pollution event in Seoul-Incheon (May 24-29, 2016). GOCI PM$_{2.5}$ inferred from the RF algorithm (background, on 6x6 km$^2$ grid scale) trained on all available data is compared to observations from the AirKorea surface network (circles).

Figure 10 shows an additional test of the RF algorithm with one of the most severe pollution events in the record, the December 16-21, 2016 Beijing winter haze episode. 24-h PM$_{2.5}$ concentrations exceeded 400 µg m$^{-3}$ at some of the network sites. While there is a tight correspondence between the GOCI and surface network 24-h PM$_{2.5}$ for Beijing grid cells (R$^2$ range: 0.74-0.99), the network observations are on average 20 µg m$^{-3}$ higher than the GOCI PM$_{2.5}$. The difference is most pronounced at the December 21 concentration peak which has mean observed value 396 µg m$^{-3}$ to the predicted 348 µg m$^{-3}$. This reflects the RF smoothing and AOD underestimate for the high tail of the distribution as previously illustrated in **Figure 4**. It nevertheless illustrates the ability of GOCI combined with our

gap-filling method to capture severe winter haze episodes that are particularly challenging to observe from space.

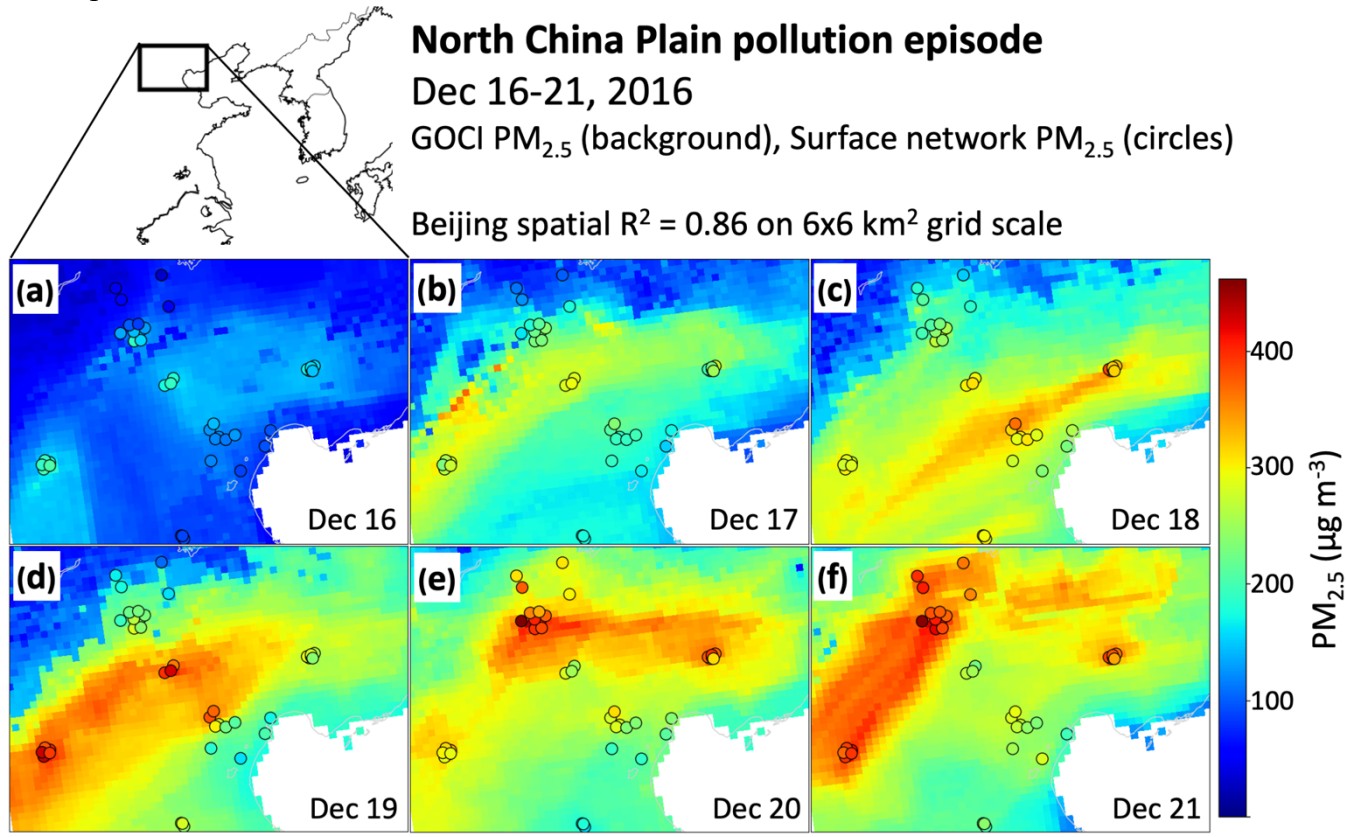

Figure 10: Same as Figure 9 but for a pollution event in Beijing on December 16-21, 2016.

### 3.4 Regional air quality model evaluation

Regional air quality model predictions of PM$_{2.5}$ are typically evaluated with observations from surface network sites, but the spatially continuous GOCI PM$_{2.5}$ fields offer more extensive coverage and hence broader opportunity for model evaluation. We demonstrate this capability here with Community Multiscale Air Quality Modeling System (CMAQ version 4.7.1) simulations for the Korean peninsula including both South and North Korea at 9-km resolution [*Bae et al.*, 2018; *Bae et al.*, 2021]. There are no surface PM$_{2.5}$ data in North Korea to train the RF so we use the South Korea categorical variable to generate the GOCI PM$_{2.5}$ fields there.

The simulation for South Korea was conducted for 2015-2019 using emissions from the Clean Air Policy Support System (CAPSS) 2016 [*Choi et al.*, 2020] for South Korea and KORUSv5 [*Woo et al.*, n.d] for outside South Korea. The simulation for North Korea was conducted for 2016 using emissions from the Comprehensive Regional Emissions Inventory for Atmospheric Transport Experiment (CREATE) 2015 [*Woo et al.*, 2020] and CAPSS 2013. Natural aerosols including sea salt

and mineral dust are included. To prepare the boundary conditions, a coarse domain at 27-km horizontal grid resolution covering Northeast Asia was used.

     **Figure 11** illustrates the increased capability for model evaluation in South Korea enabled by the GOCI PM$_{2.5}$ fields. The bottom row shows the mean 2015-2019 PM$_{2.5}$ concentrations in CMAQ compared to the AirKorea network and to GOCI PM$_{2.5}$, and the top row shows comparison scatterplots.

The top left panel compares the CMAQ simulation to 2015-2019 mean PM$_{2.5}$ observations from the 398 AirKorea network sites. The top middle panel compares the GOCI PM$_{2.5}$ to the same AirKorea network data, showing excellent agreement. The GOCI PM$_{2.5}$ fields provide 1353 points for South Korea on the 9x9 km$^2$ CMAQ grid, and the top right panel shows the resulting increase in capability for evaluation of the CMAQ simulation. It shows in particular that CMAQ underestimates PM$_{2.5}$ in coastal environments,

possibly because of unaccounted ship emissions.

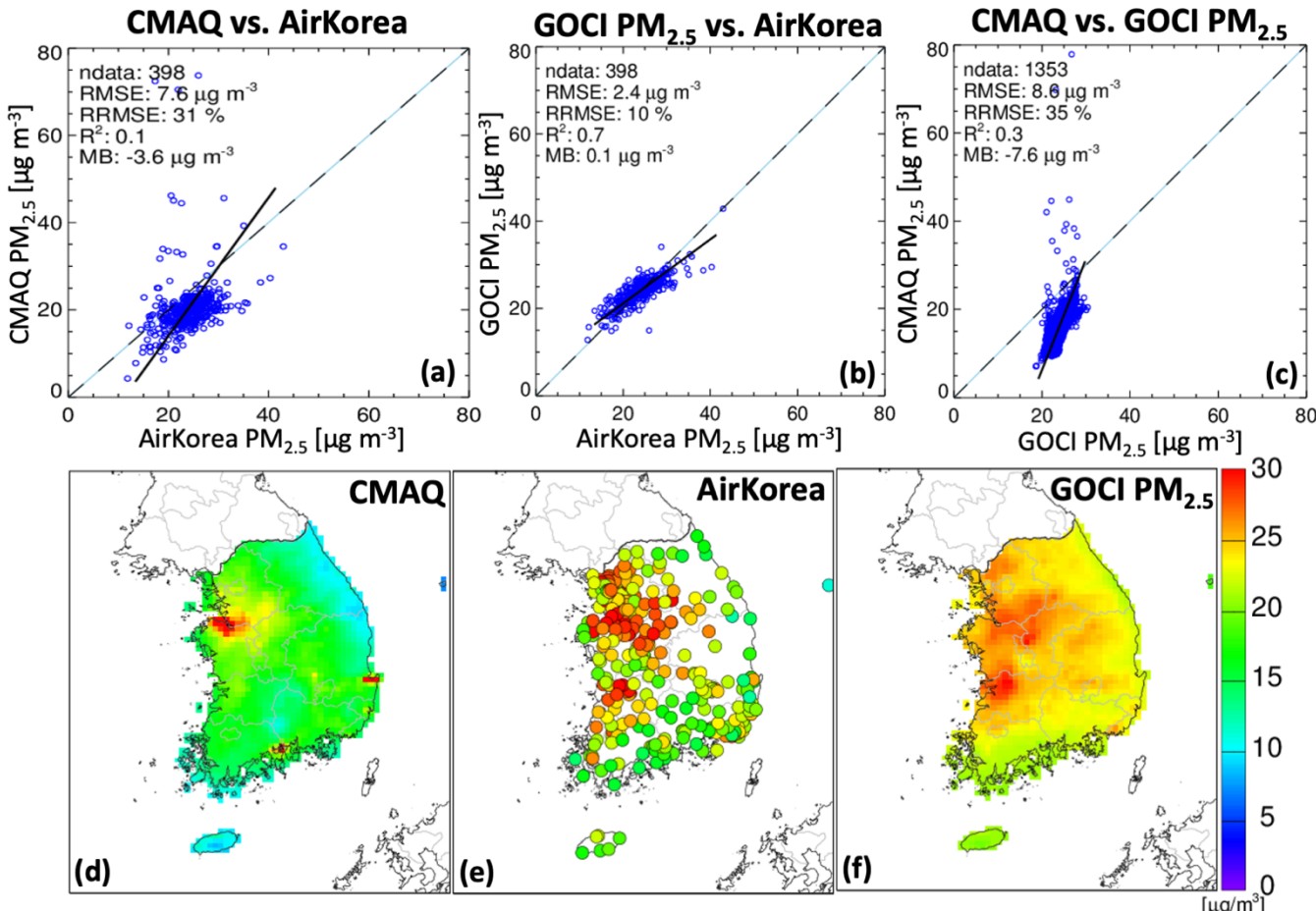

Figure 11: Mean PM$_{2.5}$ concentrations in South Korea in 2015-2019 as simulated by CMAQ, measured at the AirKorea sites, and inferred from GOCI. The top panels show scatterplots comparing the CMAQ and GOCI PM$_{2.5}$ fields to the Air Korea measurements (398 sites), and

480 CMAQ to GOCI PM$_{2.5}$ on the 9x9 km$^2$ CMAQ grid (1353 grid cells to compare). The bottom panels show maps of the mean 2015-2019 concentrations.

**Figure 12** evaluates the CMAQ simulation with the GOCI PM$_{2.5}$ fields over North Korea. Unlike in South Korea, there are no observation sites in North Korea and GOCI PM$_{2.5}$ offers the only opportunity for local evaluation. CMAQ and GOCI PM$_{2.5}$ show dramatically different patterns. The highest PM$_{2.5}$ in CMAQ is in the Pyongyang capital region, while GOCI shows highest values in the north-central region. The lack of reliable emission inventories for North Korea makes it difficult to arbitrate this difference. The RF is not trained for North Korea, which might lead to positive biases because the AOD/PM$_{2.5}$ ratio modeled in the *Zhai et al.* [2021] GEOS-Chem simulation is higher over North Korea outside the mountainous east (range: 0.010-0.013 m$^3$ µg$^{-1}$) than over South Korea (0.008-0.010 m$^3$ µg$^{-1}$). However, the difference could also be explained by missing emissions in the inventory. Further evaluation could be done with border sites in South Korea and northeastern China. China MEE sites along the border are consistent with high PM$_{2.5}$ in north-central North Korea.

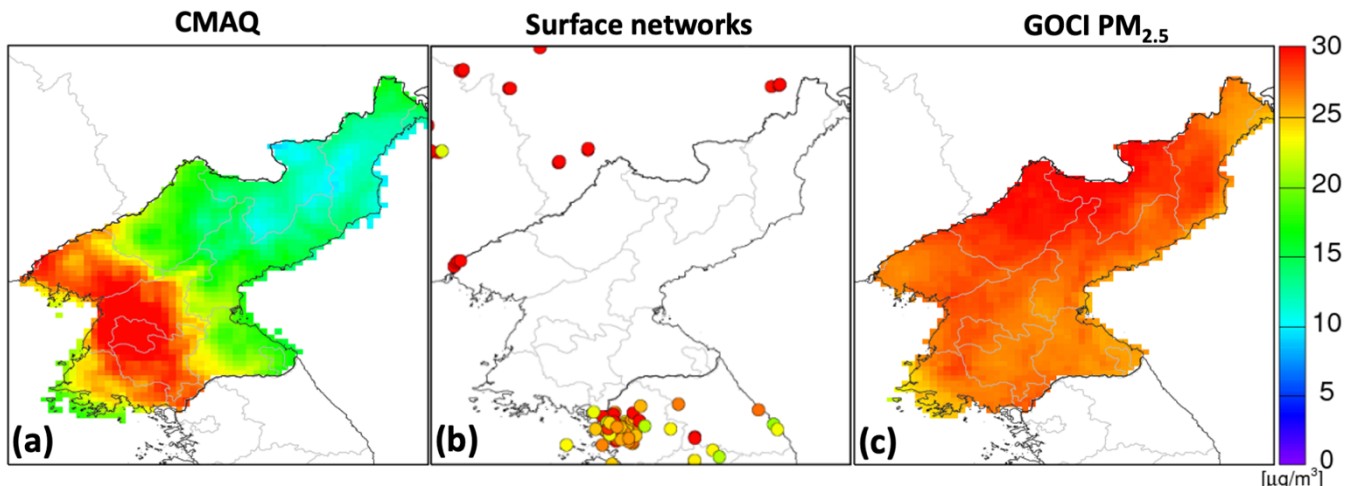

Figure 12: Mean PM$_{2.5}$ concentrations in North Korea in 2016 as simulated by CMAQ and as represented by the GOCI PM$_{2.5}$ product assuming South Korea as categorical variable. The middle panel shows surface PM$_{2.5}$ concentrations from the AirKorea and China MEE networks.

## 4 Conclusions

We used 2011-2019 geostationary aerosol optical depth (AOD) observations from the GOCI satellite instrument, in combination with a random forest (RF) machine learning algorithm trained on air quality network data, to produce a continuous 24-h PM$_{2.5}$ data set for eastern China, South Korea, and Japan at 6x6 km$^2$ resolution. The resulting gap-free GOCI PM$_{2.5}$ product complements the air quality networks that cover only 1% of 6x6 km$^2$ grid cells in eastern China, 7% in South Korea, and 8% in Japan. It provides a general dataset for PM$_{2.5}$ mapping to serve regional pollution analysis, air quality monitoring, and public health applications.

We trained the RF algorithm on gap-filled AODs from the GOCI instrument and a suite of twelve meteorological, geographical, and temporal predictor variables. Gap filling of AODs was done by a weighted combination of nearest-neighbor data and chemical transport model fields, with the weight serving as an additional predictor variable. The RF algorithm is successfully able to exploit

information encoded in AOD missingness to produce a continuous product. Testing of the RF algorithm by prediction of withheld network sites shows single-value precisions in each country of 26-32% for 24-h $PM_{2.5}$ and 12% for annual mean $PM_{2.5}$, with negligible mean bias. Accuracy statistics for $PM_{2.5}$ inferred on grid cells with no AOD retrieval (i.e., estimated using equation (1)) show similar accuracy statistics as the entire population, indicating that the gap-filling procedure does not bias the results. The

algorithm has only moderate success at predicting NAAQS exceedance events because most of these events are within the single-value precision, and also because of some smoothing of the extreme high tail of the $PM_{2.5}$ frequency distribution.

        We compared the continuous 24-h GOCI $PM_{2.5}$ fields to spatial and temporal patterns observed at the national network sites. National trends of $PM_{2.5}$ inferred from GOCI and weighted by area or

population are consistent with those observed at network sites (2015-2019 in eastern China and South Korea, 2011-2019 in Japan), confirming that the trends observed at these sites are representative. However, the network sites in eastern China and South Korea underestimate population exposure. The GOCI $PM_{2.5}$ fields over South Korea show $PM_{2.5}$ hotspots missing in the early AirKorea network (2015) that are confirmed by subsequent addition of sites to the network (2019). The spatial distribution of

GOCI $PM_{2.5}$ trends in South Korea shows decreases everywhere except in the Seoul metropolitan area where trends are flat. We show with time series in the capital cities (Beijing, Seoul, Tokyo) that the RF successfully captures the seasonality of $PM_{2.5}$ even though AOD and $PM_{2.5}$ have different and often opposite seasonalities.

        We examined the ability of the RF algorithm to map air quality on urban scales by analysis of

two multi-day pollution episodes in Seoul and Beijing. The algorithm captures the day-to-day temporal variability observed by the surface networks as well the spatial variability on the 6x6 $km^2$ scale. The highest $PM_{2.5}$ concentrations are underpredicted, which reflects the smoothing of the high tail of the distribution.

        The continuous spatial coverage of $PM_{2.5}$ provided by the GOCI fields enables improved

evaluation of the air quality models used in support of emission control policies. Comparison to a CMAQ simulation for South Korea in 2015-2019 reveals a large model underestimate in coastal environments undersampled by the AirKorea network. Comparison to a CMAQ simulation for North Korea in 2016, where the RF provides the only $PM_{2.5}$ data for model evaluation, shows drastically different patterns with the RF featuring high $PM_{2.5}$ throughout North Korea. The RF results in North

Korea could be affected by errors due to lack of training data but they appear consistent with the $PM_{2.5}$ network observations at Chinese border sites.

        More work could be done to improve our GOCI $PM_{2.5}$ product. We find in our current RF algorithm, consistent with *Hu et. al.* [2017], that the addition of certain predictor variables such as population decreases performance. This motivated our practice of excluding spatially varying but

temporally constant fields such as elevation and emissions. However, these variables have been found to be useful in other inferences of $PM_{2.5}$ from AOD data [*Kloog et al.,* 2012; *Di et al.*, 2019], so further investigation is needed on how to accommodate them in our modeling framework. A higher resolution meteorological reanalysis such as ERA5-Land [*Muñoz-Sabater et al.*, 2021] could be used for the meteorological predictor variables and enable the inclusion of additional variables such as precipitation.

Additional remote sensing products such as NDVI could also be useful. More work needs to be done to address our underestimate of the high tail of the $PM_{2.5}$ distribution, i.e., extreme pollution events. Such

an underestimate is common in RF applications [*Zhang and Lu*, 2012] but could be addressed by leveraging specialized statistical tools like extreme value theory. Additional training methods could be used to improve the ability of the RF to predict NAAQS exceedances, such as data sampling
adjustments. Moreover, it is possible that skill in modeling NAAQS exceedance could be improved by leveraging data that better captures diurnal variations of $PM_{2.5}$, as high concentrations tend to occur at night. The unique geostationary capability of GOCI to generate hourly AOD data could be used to produce an hourly $PM_{2.5}$ product. A new GOCI AOD product with 2x2 $km^2$ resolution is expected to become available in the near future and will provide motivation to explore these improvements in a new
version of our RF algorithm.

**Data availability** 24-h 6x6 $km^2$ resolution daily GOCI $PM_{2.5}$ are made freely available on DataVerse at https://doi.org/10.7910/DVN/0L3IP7.

**Author Contributions** DP and DJJ designed the study. DP developed the RF and performed analysis. SZ, MB and SK ran and analyzed chemical transport model data. SL aided in satellite data processing. JK, HL and JHK provided scientific interpretation and discussion. All authors provided input on the paper for revision before submission.

**Competing interests** The authors declare that they have no conflict of interest.

**Acknowledgements** This work was funded by the Samsung $PM_{2.5}$ Strategic Research Program and the Harvard-NUIST Joint Laboratory for Air Quality and Climate (JLAQC). GOCI data was provided by Korea Institute of Ocean Science and Technology (KIOST). DCP was funded by a US National Science
Foundation Graduate Fellowship. We thank the two anonymous reviewers for their thoughtful feedback.

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
