# Peer review of "Continuous mapping of fine particulate matter (PM2.5) air quality in East Asia at daily $6x6 \text{ km}^2$ resolution by application of a random forest algorithm to 2011-2019 GOCI geostationary satellite data"

_Atmospheric Measurement Techniques, 2021_

## Referee Comment (RC2)

[referee-annotated manuscript omitted]

---

## Author Response (AR1)

The authors try to create continuous daily PM maps from the GOCI satellite in East Ais. They only applied the well-known RF model, and I didn't see much innovation and surprise in terms of method or conclusions. In addition, we did not see any discussion and validation in AOD gap filling, which significantly reduced the reliability of follow-up work. The authors also ignored many key factors in modeling. Below are my comments and hope they are useful for improving the paper.

Response: We thank the reviewer for their helpful feedback, which has improved the paper and helped shape our plans for future work.

Specific comments:

Introduction:

Random forest is a well-known and widely used machine learning model. Please summarize the related studies on PM2.5 predictions using the RF model.

Response: The reviewer is correct to point this out, and we have added a paragraph in the introduction summarizing PM2.5 prediction using the RF model (lines 63-69).

Also, the authors are suggested to summarize recent studies on PM2.5 estimates from different geostationary satellites (e.g., GOCI, Himawari-8) since you focus on East Asia.

Response: We have added additional information and citations regarding the use of geostationary satellites to estimate PM2.5 over East Asia (lines 75-80).

Data and methods

Line 91: 8x?

Response: We removed this term and revised the text to more clearly communicate that AOD is measured every hour from 00:30-07:30 UTC and that we average this into a daily mean (lines 107-112).

Line 93-95: how about the accuracy of GOCI AOD in your study domain?

Response: We elaborated how the GOCI AOD compares favorably with AERONET with no significant variation across the study domain of East Asia (lines 115).

Line 122: Why not use the ERA5 reanalysis at a higher resolution of 0.1 degrees?

Response: The reviewer makes an excellent point. We have added a paragraph at the end of the conclusion about current limitations of this product and future work to improve it, and include a discussion of meteorology (lines 547-549).

Line 125: Several main meteorological (e.g., precipitation) and land-related variables (e.g., NDVI, DEM, …) are neglected.

In addition, the authors did not consider any humanistic factors (e.g., population, emissions) that have a large influence on PM2.5 in East Asia.

Response: The reviewer is right to bring up these datasets. We address why we do not include population within this particular modeling framework on lines 264-266 and added a mention of emissions and elevation in lines 267-268, but agree that in a future modeling paradigm should include these factors and include a discussion in the future work paragraph (lines 544-550). That discussion also includes mention of additional meteorological factors to test.

Results and discussion:

First, the authors need to show and discuss the accuracy of gap-filled AODs to ensure that they are reliable to be used for PM2.5 estimation in the next stage.

Response: We thank the reviewer for pointing out this issue. We added a paragraph discussing the ability of the gap-filled AODs, together with a length scale parameter, to accurately predict $PM_{2.5}$ without introducing new bias (lines 290-296). We also added a new supplemental figure S1 which shows predicted and observed $PM_{2.5}$ on days with and without AOD retrieval, demonstrating how our methodology recovers observed $PM_{2.5}$ distributional differences when AOD retrieval fails.

Figure 3: I am supervised that the maximum value of daily PM2.5 only is 250 μg m-3 since it can be easy to exceed > 600 μg m-3 in heavily polluted conditions in East Asia.

Response: The author is certainly correct that very high PM2.5 values occur regularly in East Asia. We modified to extend the range of the figure to 600 μg m-3 and added a comment to the caption on the 0.002% of values that were excluded even in this large range (lines 304-305). We also extended the range of the annual PM panel to incorporate all data.

Figure 5: The authors are suggested to compare and discuss their results with previous studies. Are they consistent or different?

Response: We expand our discussion of how our results compare to the literature on lines 282-287.

In addition, I also suggest adding an annual PM2.5 map for each year in your study domain.

Response: This is shown in supplemental figure S2, added in response to this comment.

Figure 6: How do your derived PM maps compare with the surface network in the other two countries?

Response: We added supplemental figures S3 and S4 to show these comparisons in China and Japan, respectively.

Figure 7: Similarly, how about PM2.5 changes in the other two countries? Please show and discuss the results.

Response: We added supplemental figure S5 to show the trends across the entire domain and briefly discuss in the main text how the North China Plain sees the largest PM2.5 improvement (lines 406-407).

Reviewer's comments on manuscript titled "Continuous mapping of fine particulate matter (PM$_{2.5}$) air quality in East Asia at daily 6×6 km$^2$ resolution by application of a random forest algorithm to 2011–2019 GOCI geostationary satellite data" by **Drew C. Pendergrass et al.**

In this paper, the authors presented results of estimating PM2.5 by using RF from the gap-filled GOCI AOD product. The objective of this paper is very clear, which intends to demonstrate the ability and also obstacles of deriving PM2.5 from satellite-derived AOD products by using machine-learning techniques. In general, the presentation of this paper is very clear and sound, however, the technique and approach used in this study have been widely used. In addition, an important aspect of this study, i.e., the gap-filled GOCI AOD, is not discussed in detail. The contents of this paper are of great importance, especially due to the crucial role of deriving surface PM2.5 concentration with continuous spatial coverage in air quality monitoring, the method used is not innovative but the results are significant and sound enough.

Response: We thank the reviewer for their comments. We also have added new material expanding on our approach to use information encoded in AOD missingness in the introduction (lines 88-89; 93-97), the innovativeness of which we believe was not sufficiently clear in the original draft.

Comments/suggestions:

1. Line 263-265. This statement seems does not have data to backup. it is strongly suggested to include the data/figures. And, the gap-filled GOCI AOD is a crucial base for deriving a continuous map of PM$_5$ concentration in this study. It really deserves a paragraph to describe how well the gap-filled GOCI AOD performed.

Response: We thank the reviewer for pointing out this issue. As noted in a response to the first reviewer, we added a paragraph discussing the ability of the gap-filled AODs, together with a length scale parameter, to accurately predict PM$_{2.5}$ without introducing new bias (lines 290-296). We also added a new supplemental figure S1 which shows predicted and observed PM$_{2.5}$ on days with and without AOD retrieval, demonstrating how our methodology recovers observed PM$_{2.5}$ distributional differences when AOD retrieval fails.

2. Line 270. The labels in these figures need to be redone. It is suggested to add "a" and "b" respectively for each figure.

Response: We have added letter labels to all figures in the paper and modified captions accordingly.

3. Line 350. The labels in these figures are confusing. Labels are added only in the third figure, but lines in the other two figures have different colors, it is suggested to add labels to other figures as well, Also, explain them in the caption.

Response: We have added color coded legends to all panels of figures 5 and 8 and modified captions to clarify the terms we use in the legends.

4. Line 290-310. The challenge in predicting NAAQS exceedances is well presented in this paragraph. And the authors mentioned that several attempts were made but no improvement was seen. Part of the problem could be non-equal sampling in the different PM2.5 ranges, it is suggested to try to train RF in training datasets that have a roughly equal- number of samples in different PM2.5 ranges. Secondly, GOCI AOD performance has been well documented in the validation studies. What about carrying out bias correction for GOCI AOD for different ranges of AOD before training?

Response: The reviewer is correct to point out that more could be attempted to resolve the NAAQS prediction issue. We have added a discussion of what we might try in future work to lines 553-557.